# Pre- and Post-Operative Education and Health-Related Quality of Life for Patients with Hip/Knee Replacement and Hip Fracture

**DOI:** 10.3390/healthcare11030329

**Published:** 2023-01-22

**Authors:** Yen-Mou Lu, Je-Ken Chang, Pin-Yu Lin, Yi-Jing Lue

**Affiliations:** 1Department of Orthopaedics, School of Medicine, College of Medicine, Kaohsiung Medical University, Kaohsiung 807, Taiwan; 2Division of Pediatric and Spinal Orthopedics, Department of Orthopedics, Kaohsiung Medical University Hospital, Kaohsiung 807, Taiwan; 3Department of Sport Medicine, Kaohsiung Medical University Hospital, Kaohsiung 807, Taiwan; 4Department of Orthopaedics, Kaohsiung Medical University Hospital, Kaohsiung 807, Taiwan; 5Department of Orthopaedics, Kaohsiung Veterans General Hospital, Kaohsiung 813, Taiwan; 6Department of Physical Therapy, College of Health Sciences, Kaohsiung Medical University, Kaohsiung 807, Taiwan; 7Department of Medical Research, Kaohsiung Medical University Hospital, Kaohsiung 807, Taiwan

**Keywords:** arthroplasty, fracture, education, quality of life

## Abstract

Arthroplasty for the hip/knee and surgeries for hip fractures are increasing worldwide. The aims of this study were to investigate changes in health-related quality of life (HRQOL) after surgery with an early mobility education program, and to explore their associations with pain and anxiety. Pain intensity and anxiety were assessed with the visual analogue scale (VAS) and Beck Anxiety Inventory (BAI), and HRQOL was assessed with the Short Form-36 (SF-36). The physical component summary (PCS) and mental component summary (MCS) and eight subscales of the SF-36 were calculated. At pre-operation, the patients suffered from moderate pain and mild anxiety, and their HRQOL scores were low (4.9, 7.8, 35.4, and 48.2 for the VAS, BAI, PCS, and MCS, respectively). The pain, anxiety, and HRQOL improved after surgery and had moderate to large effect sizes at 6-month follow-up (Glass’s delta = 1.23, 0.88, 0.81, and 0.67 for VAS, BAI, PCS, and MCS, respectively). Pain and anxiety were strongly correlated to HRQOL at each stage, with the maximum correlation (r = −0.34 to −0.93) reached at 6-month follow-up. The surgery effectively improves HRQOL, as the reduced pain and anxiety lead to better physical and mental HRQOL.

## 1. Introduction

The prevalence of arthroplasty for hip and knee joints is increasing worldwide, and the prevalence rises with age. At the end of 2016, for the age groups of 65 to 84 years and over 85 years, the overall prevalences were 13.3% and 22.5%, and about 3.4% of the total population had at least one joint replacement, as reported by a study in Australia [1]. In a 2010 prevalence rate study of the total US population, an estimated 2.5 and 4.7 million individuals had total hip replacement (THR) or total knee replacement (TKR), and the trends show a rapid rise in prevalence over time, with more people in younger age groups needing joint replacement [2]. In Taiwan, from 1998 to 2009, the number of primary THRs increased from 3726 to 4972 (a 33% increase), and that of primary TKRs rose even more (from 6062 to 14,467, a 138% increase) [3]. The projected increases in primary THRs and TKRs by 2030 are 69.7% and 508.2% relative to the respective rates in 2005. A cross-national comparison study also showed that the median time to a primary TKR or THR was about half as long in the US as in the UK, and the primary replacement rate for the hip was higher in the US (1.19 per 100 person-years) than in the UK (0.76 per 100 person-years) [4]. The hip fracture prevalence rate is also quickly rising; the global number of hip fractures is predicted to increase from 1.26 million in 1990 to 4.5 million by the year 2050 [5]. Hip fracture is a debilitating condition associated with a long period of hospitalization, poor quality of life (QOL), more social costs, disability, and mortality [5]. Therefore, many people need arthroplasty or surgery, and an essential issue for the world is how to promote function after surgery.

Degenerative arthritis and osteoporosis are common diseases in the lower extremities of the elderly. Pain due to arthritis and inactivity can significantly interfere with daily and social activities. Osteoporosis can easily lead to femoral neck fracture. Although fracture is not difficult to cure, patients commonly stay in bed for an extended period, which consequently induces many complications. Early activity after surgery needs to include education for patients with hip fracture and arthroplasty, such as getting out of bed as soon as possible and following a specific exercise plan. Furthermore, an education program focused on early activity could conserve medical resources and reduce the social burden.

QOL is considered one of the main postoperative outcomes, and after arthroplasty, most patients report improved QOL [6]. However, some are not satisfied [7]. Many factors, such as pre-operative psychological factors and other patient-related variables, influence the outcomes. Anxiety and feelings of a loss of control are common around surgery. Excessive anxiety and depression may affect physical recovery, reduce QOL, prolong hospital stays, and increase the costs of treatment [8]. Good understanding of the operation and postoperative routines due to education can reduce patient anxiety, which in turn may improve the patient’s ability to cope with postoperative pain [9]. In a systematic study, pre-operative anxiety, pain, and poor function were the most significant predictors of poor health-related quality of life (HRQOL) [10]. Pre-operative anxiety and depression were also the most common preoperative predictors of dissatisfaction [7]. The prevalence of pre-operative anxiety/depressive symptoms was high, and patients with anxiety/depressive symptoms had worse patient-reported outcomes and less satisfaction than patients without such symptoms [11].

Although current evidence clearly shows that pre-operative pain and anxiety affect HRQOL, few studies explored changes in pain and anxiety during the subsequent stages, and no information is available on concurrent pain and anxiety associated with HRQOL in the follow-up stages. The aim of this study therefore was to investigate HRQOL at different stages and the associations of pain and anxiety at the same stages by providing an early activity education program for patients with TKR, THR, and hip surgery.

## 2. Materials and Methods

### 2.1. Participants

A convenience sample of patients was recruited from the Department of Orthopedics of Kaohsiung Medical University Hospital. Patients were included if they met the following criteria: age of 18 years or older; scheduled THR or TKR surgery at the hospital or surgery due to hip fracture; the ability to read traditional Chinese; and the absence of physical limitations affecting the ability to complete the self-administered questionnaire. The exclusion criteria included history of knee or hip surgeries, other medical diseases, and psychiatric history. The primary diagnoses for arthroplasty included osteoarthritis, rheumatoid arthritis, traumatic arthritis, and avascular necrosis. After explanation of the research program, written informed consent was obtained from all patients. We expected the effect sizes to be large (0.8) and moderate (0.5) for physical and mental quality of life, respectively, between 6-month follow-up and pre-operation. The sample sizes were estimated with a two-sided significance of 0.05 and a power of 0.8; a total of 15–34 patients was required (calculation by G*Power 3.1.9.7). Considering the attrition effect of long-term follow-up, the total sample size in the study was 100.

### 2.2. Procedure

After receiving an explanation of the research program, each patient was asked to complete a questionnaire booklet containing the visual analog scale (VAS) for pain severity, the Beck Anxiety Inventory (BAI) for anxiety, the MOS Short Form 36 (SF-36) for HRQOL, and demographic questions. Each patient was tested on their knowledge of arthroplasty or hip surgery before the education was provided. The test had 14 items with a score range of 0–14 (0 indicating poor knowledge). The tests for TKR, THR, and hip fracture were different. A one-on-one preoperative education program was provided before and after the surgery by a physiotherapist or nurse; a booklet and a video were used for education. At the discharge, the test of the knowledge of arthroplasty or hip surgery, VAS, and BAI were administered again, and the physiotherapist or nurse rechecked whether the patient well understood the education program. At 6 weeks and 6 months post-operation, we first contacted the patient by telephone and then mailed the questionnaire booklet to him/her. The booklet included the VAS, BAI, and SF-36 questionnaires.

### 2.3. Educational Programs

The purpose of the education project was to improve the early activities of patients with TKR, THR, and hip fracture. The project was reviewed and approved by the Health Promotion Administration, Ministry of Health and Welfare, in Taiwan. The details of the education program were developed by a multidisciplinary team including two orthopaedic physicians, three physiotherapists, an occupational therapist, a psychotherapist, a nutritionist, a social worker, and a nurse. We developed three booklets for patients with TKR, THR, and hip fracture, respectively. We developed videos in two spoken languages, Mandarin and Taiwanese; therefore, in all, six videos were designed for patients with TKR, THR, and hip fracture. The content of the booklets included the progress of a person undergoing hip or knee surgery, from home preparation to hospital stay; information on anatomy, surgery, and the prothesis for replacement; the clinical pathway; precautions following surgery; postoperative recovery; positioning; early activities; daily living activities; exercise; advice on diet, social services, and planning for discharge; and the outpatient clinic program. The video focused on the precautions and positions after surgery, exercise training, early activity training (such as transfer from lying to sitting and sitting to standing, walking with assistive devices, and ascending/descending stairs), and daily living activity training with equipment. The educational program was delivered by a trained physiotherapist and a nurse. The content of the booklet was verbally explained before the surgery. One day after the surgery, the patient watched the video and then was taught the exercises and activities through demonstration and practice.

### 2.4. Instrument

#### 2.4.1. VAS

The pain intensity was assessed with the visual analogue scale (VAS). The VAS is a 10 cm horizontal line with “no pain” written at the left endpoint and “severe pain” written at the right endpoint [12]. The patient was asked to draw a vertical line to mark a point corresponding to the magnitude of his/her current pain. Cut points were recommended as follows: ≤3.4, mild pain; 3.5 to 7.4, moderate pain; and ≥7.5, severe pain [13].

#### 2.4.2. BAI

The Beck Anxiety Inventory (BAI) is a 21-item self-report questionnaire measuring symptoms of anxiety [14]. The BAI scores are classified as minimal anxiety (0 to 7), mild anxiety (8 to 15), moderate anxiety (16 to 25), and severe anxiety (30 to 63) [15]. The BAI was shown to better differentiate anxiety from depression than does the State-Trait Anxiety Inventory [16].

#### 2.4.3. SF-36

HRQOL was measured with the Medical Outcomes Study Short Form 36 (SF-36). The SF-36 has eight subscales: physical functioning (PF), role limitations due to physical health problems (role–physical, RP), bodily pain (BP), general health (GH), vitality (VT), social functioning (SF), role limitation due to emotional problems (role–emotional, RE), and mental health (MH) [17]. The transformed scores of each subscale range from 0 to 100, with higher scores indicating better health QOL. Two summary measures, the physical component summary (PCS) and mental component summary (MCS), were calculated from these eight subscales to demonstrate the overall physical and mental function, respectively [17]. The Taiwan version of the SF-36 was shown to have good reliability and validity [18]. The normative values were derived from the 2001 Health Interview Survey of 17,515 subjects and are available with age and sex stratification [19]. The subscale scores of the patients were compared with the normative values of the SF-36 Taiwan version.

### 2.5. Data Management and Statistical Analyses

For HRQOL, to determine the magnitude of deviation of the eight subscales of the patients from those of the general population, age-/sex-matched normative values were selected and coded according to the age and sex of each patient. The study stages included the pre-operation stage, discharge (without SF-36 data), post-operation 6-week follow-up, and post-operation 6-month follow-up. One-way analysis of variance (ANOVA) with post hoc analysis was used to analyze the difference among stages. The least significant difference (LSD) was used for post hoc analysis. The magnitude of change in scores was evaluated by calculating the effect size (ES). Glass’s ∆ (delta) was used to calculate the ES because the experimental manipulation potentially affected the standard deviation [20]. The ES was computed as the difference between the pre-operation and 6-month follow-up scores divided by the standard deviation of the pre-operation scores. An ES can be considered large if the value is greater than 0.8, moderate if the range is between 0.5 and 0.79, and small if the range is between 0.2 and 0.49, according to Cohen [21,22]. The associations between VAS/BAI and HRQOL (two summary measures and eight subscales of the SF-36) were tested with Pearson’s correlation coefficients. Correlation coefficients between 0.3 and 0.59 indicated moderate correlation, and those of 0.6 or greater, high correlation [23]. The α level was set at 0.05. Statistical analyses were performed in SPSS for Windows, release 20.0.

## 3. Results

### 3.1. Participant Demographics

The demographic data and clinical characteristics of the patients pre-operation are shown in Table 1. A total of 100 patients (mean age = 64.8 years old) participated in this study. Females (*n* = 64) outnumbered males (*n* = 36); most of them received arthroplasty (TKR, *n* = 52; THR, *n* = 31) and 17 patients received hip surgery to treat fracture. The pain of the patients was moderate (mean VAS near to 5) and their anxiety was mild (mean BAI near to 8) before the operation. Before the operation, the scores of the SF-36 were low in both physical QOL (PCS = 35.4) and mental QOL (MCS = 48.2). The scores of the VAS, BAI, PCS, and MCS were not significantly different among subgroups (patients with THR, TKR, and hip fracture) before the operation. The mean test scores for the patients’ knowledge were 8.2 ± 2.5 at the beginning of pre-education, and these scores improved to 11.7 ± 1.9 on the day of discharge (*p* < 0.001). The test scores at the beginning and discharge were not significantly different among subgroups. The mean hospital stay was 6.5 ± 1.8 days. The patients with hip fracture stayed about one day longer (7.4 ± 3.2) than the patients with THR/TKR (6.1 ± 1.4 and 6.5 ± 1.3, respectively) did; however, the difference was not significant (*p* = 0.074).

### 3.2. Health-Related Quality of Life (HRQOL)

The scores of the two summary measures at the three stages (pre-operation, 6-week, and 6-month follow-up) and scores of the eight subscales of the age-/sex-matched normative values and the patients at the three stages are listed in Table 2. Two summary measures and many subscales indicated significant differences among the stages. The effect sizes of the PCS, PF, BP, and RE, which were compared between 6 months and pre-operation, were large (∆ *=* 0.81, 1.02, 1.21, and 0.96), indicating great improvements in HRQOL 6 months after the operation. The patients with THR had the greatest PCS effect sizes (∆ *=* 1.3), while those with TKR/fracture showed no significant improvement. At the 6-week follow-up, the PCS scores of the patients with THR (41.9 ± 9.1) were higher than those of the patients with TKR and hip fracture (36.8 ± 8.1 and 34.4 ± 7.1, respectively; *p* < 0.05). At the 6-month follow-up, the scores of the PCS were not significantly different among subgroups. At both 6-week and 6-month follow-ups, the scores of the MCS were not significantly different among subgroups. The effect size of the MCS was moderate (∆ *=* 0.67) when compared between 6 months and pre-operation. The patients with TKR had larger MCS effect sizes (∆ *=* 0.73) than the patients with THR did (∆ *=* 0.66).

In the pre-operation stage, five of the eight subscales were lower than the age-/sex-matched normative values (*p* < 0.001). Figure 1 displays the results of five of the eight subscales at the three stages and compares them with the scores of the age-/sex-matched normative values. In the 6-month post-operation stage, PF and RP were still lower than the age-/sex-matched normative values (*p* < 0.05); however, RE and MH were higher than those normative values (*p* < 0.05). In the subgroup comparisons, only the SF subscale at the pre-operation stage was significantly different; the scores of the patients with TKR were higher (81.6 ± 21.4) than those of the patients with THR and hip fracture (67.9 ± 29.7 and 65.4 ± 25.6; *p =* 0.019 and 0.023, respectively).

### 3.3. Pain/Anxiety and Associations with HRQOL

The scores of the VAS and BAI were significantly lower at discharge than at the pre-operation stage (VAS changed from 4.9 to 3.0, *p* ≤ 0.001, and BAI changed from 7.8 to 5.2, *p* ≤ 0.001, respectively) (Table 3). The changes in the VAS were large between discharge and 6-week follow-up, decreasing by nearly 50% (from 3.0 to 1.7, *p* ≤ 0.001). Among subgroups, the patients with TKR reported more pain (VAS *=* 2.2 ± 2.6) than the patients with THR (0.8 ± 1.5, *p =* 0.046) at 6-week follow-up. BAI scores decreased by over 60% at 6-month follow-up as compared with discharge (from 5.2 to 1.9, *p* ≤ 0.001). The effect sizes of the VAS and BAI, which were compared between 6 months and pre-operation, were large (∆ *=* 1.23 and 0.88), indicating great improvements in pain and anxiety 6 months after the operation. Among subgroups, moderate to large effect sizes could be found, but the VAS had no significant improvements in patients with fracture (Table 3). For all stages, the scores of the BAI were not significantly different among subgroups. Briefly, both the VAS and BAI had large effect sizes. The correlation coefficients between the VAS and BAI were 0.28, 0.48, 0.58, and 0.51 for pre-operation, discharge, 6-week follow-up, and 6-month follow-up, respectively.

In each stage, the VAS scores correlated well with the PCS (r *=* −0.34, −0.46, and −0.69), and BAI scores correlated well with the MCS (r *=* −0.43, −0.55, and −0.74) (Table 4). At the preoperative stage, VAS scores were moderately correlated with the PCS and BAI scores were moderately correlated with the MCS. The correlations were strong at 6 weeks and even stronger at 6 months. At 6 months, VAS scores were highly correlated not only with the PCS (r *=* −0.69), but also with the MCS (r *=* −0.70). Similarly, at 6 months, BAI scores were highly correlated with the MCS (r *=* −0.74) and moderately correlated with the PCS (r *=* −0.46). Preoperative PCS scores were also moderately correlated with the 6-month follow-up VAS scores (r *=* −0.52). On the other hand, preoperative MCS scores were not correlated with the BAI scores at 6-month follow-up. For the subscales, VAS scores were moderately to highly correlated with the subscales (r from −0.56 to −0.93), and BAI scores were slightly lower but also moderately to highly correlated with the subscales (r from −0.34 to −0.74) at 6-month follow-up.

## 4. Discussion

At the pre-operative stage, the patients had moderate pain, mild anxiety, and poor HRQOL. Over half of the subscales of the SF-36 had poor scores as compared with the normative values. After education and arthroplasty or hip surgery, the patients improved quickly, the average hospital day was short, and the pain and anxiety significantly decreased at discharge. Most of the subscale scores of the SF-36 continued to improve. The effect sizes between 6-month follow-up and pre-operation were large, in both the physical (physical functioning and bodily pain) and mental (role emotional and mental health) domains. The scores of role emotional and mental health were even significantly better than the normative values. For each stage, the physical summary measure was correlated with pain severity, and the mental summary measure was correlated with anxiety. The associations (between PCS and VAS, and between MCS and BAI) changed from moderate at pre-operation to high at 6-month follow-up.

### 4.1. Education Program

The surgery and education program decreased the pain and anxiety of the patients with TKR, THR, and hip fracture. The patient responses to the education programs were good. Especially after watching the video, patients immediately practiced the activities and exercises taught and demonstrated by the physiotherapist or nurse. We encouraged the patients to strive for early mobility and to do the exercises every day, as well as to learn the techniques for daily activities with or without assistive devices. The average hospital stay was only 6.5 days. A nationwide, population-based study from Taiwan showed that, from 2001 to 2005, the average lengths of stay were 8.4 and 8.9 days for THR due to osteoarthritis and rheumatoid arthritis, respectively, and 8.4 and 8.2 days for TKR due to osteoarthritis and rheumatoid arthritis, respectively [24]. The mean ages of the population-based studies on THR and TKR were about 60 and 68 years, respectively. The education program was not difficult for the older group (age = 80–94 years) in this study. The average test scores improved from 7.5 to 11.0, which was only slightly lower than the average test scores (from 8.2 to 11.7 for all patients). A meta-analysis concluded that the length of hospital stay for preoperative education was significantly reduced by about two days on knee replacement but smaller and non-significant on hip replacement [25]. Reduction in hospital stays may be influenced by many factors, such as the clinical pathway, treatments by different surgeons or with different techniques, and the condition of the patients. Our study had no control group due to ethical considerations; therefore, we are unable to provide a comparison of the effects with or without education.

Preoperative education resulted in a 5.1-point reduction (on the 60-point scale) in preoperative anxiety as compared with usual care (calculated from four trials) in McDonald’s meta-analysis [25]. Few studies examined pain severity or anxiety at discharge, so the findings of our study could provide a profile of the pain and anxiety of patients at discharge. The pain severity, tested by VAS at discharge, decreased by about 40% (from 4.9 to 3.0), and the anxiety, tested by BAI, decreased by about 33.3% (from 7.8 to 5.2). The VAS and BAI had a stronger association at discharge than at pre-operation; the correlation coefficient *r* changed from 0.28 to 0.48.

### 4.2. The Change in HRQOL after Surgery

Both physical and mental HRQOL improved, with moderate to large ESs at 6 months post-operation. The ES of the PCS was larger than that of the MCS, and the ESs of many subscales were large, such as those of the physical functioning, bodily pain, and role emotional subscales (∆ = 1.02, 1.21, and 0.96, respectively) (Table 2). Among the subgroups, the improvements in PCS were different. The patients with THR had the largest ES, while the patients with TKR/fracture did not significantly improve at 6-month follow-up. In clinical practice, we need to pay more attention to physical training to improve patients’ abilities to engage in daily living activities so as to improve their quality of life after surgery, especially in the case of patients with hip fracture. The physical functioning and role physical were the two lowest subscales (score = 36.4 and 22.3) at pre-operation. The physical functioning improved, but not significantly, at 6-week follow-up, and both physical functioning and role physical significantly improved at 6 months (score = 60.5 and 40.7). Compared with the age-/sex-matched normative values, the scores of the physical functioning and role physical at 6 months were still low (75.4 vs. 60.5 and 62.9 vs. 40.7 for physical functioning and role physical, respectively).

To our best knowledge, only one study investigated the HRQOL of knee arthroplasty at 6 weeks post-operation, but not much change in the subscales was found in that study [26]. Our study showed that the PCS scores of patients with THR improved more (from 34.5 to 41.9) than those of patients with TKR (from 35.4 to 36.8) and hip fracture surgery (from 37.1 to 34.4) at 6 weeks post-operation (Table 2). The pain of the patients with THR also decreased (from 7.7 to 0.8) more than that of patients with TKR (5.1 to 2.2) and hip fracture for surgery (from 4.5 to 1.8) at 6 weeks post-operation (Table 3). Most previous research on HRQOL for arthroplasty was designed for longer follow-up periods, such as 6 months, 1 year, 2 years, mid-term follow-up (equal or over 3 years), and long-term follow-up (up to 10 years or more). The scores of the eight subscales before the operation were similar to the scores of the eight subscales reported in our study [27,28,29,30]. Those patients with TKR/THR reported poor physical and mental QOL. For example, in Shi et al.’s study, the patients had low physical functioning and very low role physical scores (score = 12) at pre-operation; however, the role physical score improved greatly to 48 at 6 months post-operation [27]. The bodily pain score in our study improved from 44.9 to 76.6, while that in Shi et al.’s study improved only slightly, from 42.3 to 48.4, at 6 months post-operation. The bodily pain subscale is based on only two items. According to the FDA, patient-reported outcome data should be collected directly from patients, and individual items may also reveal useful information about individual patients in clinical practice. The first item was patient self-rated health status, which could be used as an indicator of the general health condition of the patient. In Busija’s study, they investigated the magnitude of change in SF-36 scores in four types of orthopedic surgery; the eight subscales scores were similar in patients with TKR and THR, and the results of physical functioning, role physical, and bodily pain significantly improved, with large ESs, at 6 months post-operation [28].

A five-year follow-up study for THR showed that the greatest improvement was found at the six-month assessment [31]. We suggest that 6 months post-operation is a good time point for assessment of HRQOL, as the patients recover quickly and their scores may not too far from the optimal level. The scores of the eight subscales at mid-term follow-up, or 3 years or longer, showed little improvement [29,30,31,32,33]. For mental health, our study found that the role emotional and mental health subscales were low at pre-operation, but the poor QOL improved quickly, and surprisingly, the scores exceeded the norms at 6 months post-operation in this study. The SF-36 can be used to show physical and mental changes in groups and can be used for comparison with population norms [28].

### 4.3. Pain/Anxiety and Association with HRQOL

The associations between pain and HRQOL, as well as between anxiety and HRQOL, increased at different stages of assessment. Pain severity was correlated with physical HRQOL, and anxiety was correlated with mental HRQOL (Table 3). This association was reported between the scores on the PCS and MCS at 6 weeks and 6 months post-operation, with anxiety and depression measured by the Hospital Anxiety and Depression Scale [26]. That study did not present correlation coefficients; therefore, the magnitude of the association could not be compared with our findings.

Most previous research focused on pre-operative anxiety as a predictor of a poorer outcome and poor QOL [9]. For example, Duivenvoorden et al. showed that the prevalence of anxiety was high at pre-operation and at postoperative follow-ups (3 and 12 months), but the prevalence decreased in both hip and knee patients with arthroplasty, and pre-operative anxiety significantly predicted a worse QOL [11]. Negative emotions, such as anxiety and depression, affected postoperative scores on evaluations of functional knee recovery and QOL after TKR [34].

We also found that, at 6 weeks, the BAI was also moderately correlated with the PCS, and at 6 months, the VAS was also highly correlated with MCS and the BAI was still moderately correlated with the PCS. Therefore, the VAS and BAI were correlated with both the PCS and MCS at 6 months. Furthermore, the correlation became very strong. The possible reasons may be that the pain and anxiety of most of the patients decreased at 6 months (1.2 and 1.9, respectively), while the standard deviations of the pain and anxiety became relatively large (2.5 and 3.7, respectively) (Table 3). Briefly, the patients had greatly diverse outcomes in terms of pain and anxiety; some still had severe pain and/or anxiety at 6 months post-operation, which potentially influenced their physical and mental HRQOL. All subscales of the SF-36 were moderately to highly correlated with VAS and BAI scores at 6 months. The clinical implication is that healthcare providers treating similar populations need to care about the pain and anxiety at each stage of follow-up, as pain and anxiety are strongly correlated with HRQOL. Decrease in pain and anxiety is an essential issue for clinical practice and for patients to feel comfortable at pre-operation and follow-up.

### 4.4. Limitations

The limitations of our study should be noted. The first is the generalizability of our findings. The study sample was a small number of patients with arthroplasty and hip fracture recruited from a single medical center. Multicenter studies are needed to improve the generalizability. We provided the norms (an age- and sex-matched group) to precisely reveal the conditions of our sample. The second is the follow-up period. This study explored only the short-term follow-up and could not provide information on mid-term and long-term follow-up. The third is the lack of a control group. The study was not a randomized trial design; therefore, some questions cannot be answered. Finally, many factors may be associated with HRQOL. However, this study assessed pain and anxiety at each stage and found high correlations with HRQOL. Other factors may also be associated with HRQOL and need to be studied in the future.

### 4.5. Conclusions

In conclusion, education programs focused on early mobility were delivered by booklet and video for patients with knee/hip arthroplasty or hip fracture surgery. The patients had moderate pain and mild anxiety at pre-operation; after surgery, the pain and anxiety obviously decreased at 6 weeks and 6 months, respectively. Both physical and mental HRQOL were poor as compared with those of age- and sex-matched norms at pre-operation. The ES of the physical functioning subscale was large at 6 months, but the scores did not reach the normative values. The ESs of the role emotional and mental health subscales were large and moderate at 6 months, and the scores were even higher than the normative values. Pain and anxiety were highly and moderately correlated with physical and mental HRQOL. It is important for healthcare providers to consider pain and anxiety so as to improve the quality of life of patients with arthroplasty and hip surgery.

## Figures and Tables

**Figure 1 healthcare-11-00329-f001:**
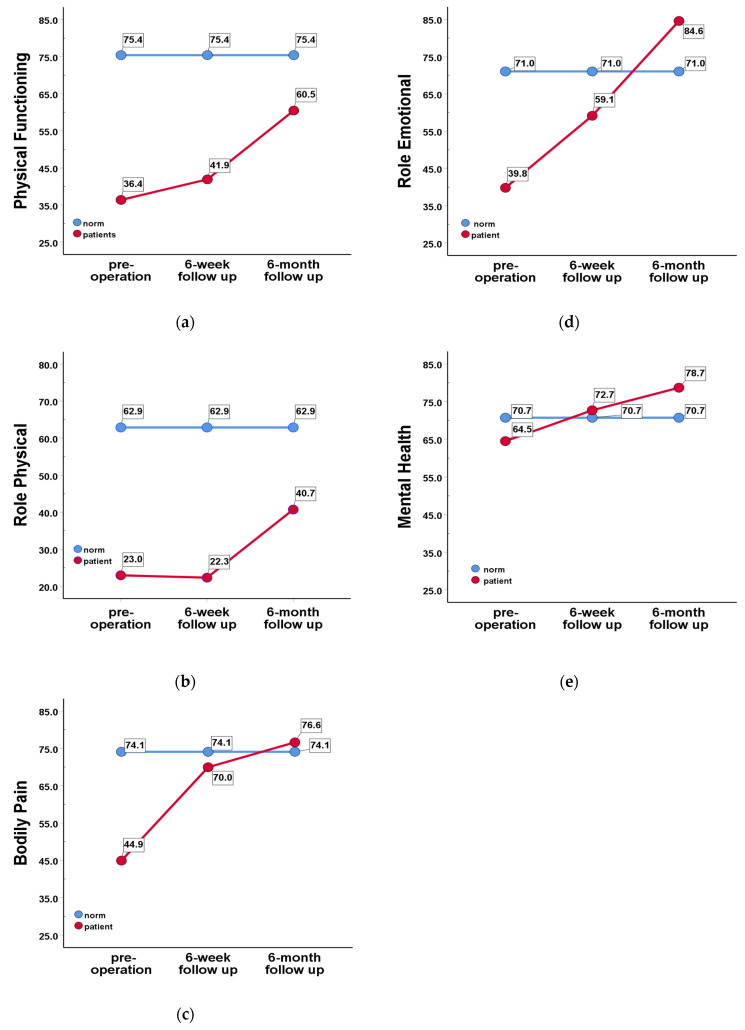
The SF-36 subscales at the pre-operation, 6-week follow-up and 6-month follow-up stages, and age-/sex-matched normative values: (**a**) physical functioning subscale; (**b**) role limitations due to physical health problems subscale; (**c**) bodily pain subscale; (**d**) role limitations due to emotional problems subscale; and (**e**) mental health subscale.

**Table 1 healthcare-11-00329-t001:** Demographic data and clinical characteristics of patients before operation.

	Mean (SD)/Range
Age (years, number)	64.8 (13.9)/18–94
(<65 years, 41; 65–79 years, 49; 80–94 years, 10)	
THR/TKR/Fracture	56.3 (15.1)/70.3 (6.7)/63.5 (19.7)
VAS	4.9 (3.0)/0–10
THR/TKR/Fracture	4.7 (3.1)/5.1 (3.0)/4.5 (3.0)
BAI	7.8 (6.7)/0–37
THR/TKR/Fracture	7.5 (6.2)/7.2 (5.7)/10.3 (10.0)
Test score	8.2 (2.5)/0–13
THR/TKR/Fracture	8.2 (2.9)/8.0 (2.2)/8.9 (2.6)
SF-36	
PCS	35.4 (8.1)/18.2–58.7
THR/TKR/Fracture	34.5 (8.4)/35.4 (6.6)/37.1 (11.6)
MCS	48.2 (10.4)/27.8–71.8
THR/TKR/Fracture	47.5 (11.6)/48.7 (10.0)/48.2 (10.0)
Diagnosis	Number
THR	31
TKR	52
Hip fracture	17
	Total (THR/TKR/Fracture)
Sex	
Male	36 (15/13/8)
Female	64 (16/39/9)
Marital status	
Single/widowed	17 (4/9/4)
Married	83 (27/43/13)
Education	
Elementary school	67 (14/43/10)
High school	24 (11/7/6)
University	9 (6/2/1)
Work status	
Retired or without work	73 (22/42/9)
Work	21 (9/7/5)
Student	6 (0/3/3)

VAS, visual analogue scale; BAI, Beck Anxiety Inventory; PCS, physical component summary; and MCS, mental component summary.

**Table 2 healthcare-11-00329-t002:** The SF-36 summary measures and subscales at pre-operation, and 6-week and 6-month follow-up of the age-/sex-matched normative values and patients.

	Norm (*n* = 100)	Pre-Operation (*n* = 100)	6 Weeks (*n* = 84)	6 Months (*n* = 55)	*p*	ES
Summary Measures		Mean (SD)				
PCS	--	35.4 (8.1)	38.2 (8.7)	42.0 (11.0) ^#,§^	<0.001	0.81
THR		34.5 (8.4)	41.9 (9.1) ^#^	45.4 (11.4) ^#^	<0.001	1.30
TKR		35.4 (6.5)	36.8 (8.1) *	39.4 (8.7)	0.11	0.62
Fracture		37.1 (11.6)	34.4 (7.1) **	37.9 (17.5)	0.75	0.07
MCS	--	48.2 (10.4)	51.8 (12.4) ^#^	55.2 (9.6) ^#,§^	<0.001	0.67
THR		47.5 (11.6)	48.4 (13.5)	55.2 (7.7) ^#,§^	0.04	0.66
TKR		48.7 (10.0)	54.4 (11.6) ^#^	56.0 (10.4) ^#^	0.009	0.73
Fracture		48.2 (10.0)	51.8 (11.0)	51.3 (15.4)	0.67	0.31
Subscales						
PF	75.4 (14.5) ^†,‡,&^	36.4 (23.7)	41.9 (22.9)	60.5 (24.5) ^#,§^	<0.001	1.02
RP	62.9 (17.5) ^†,‡,&^	23.0 (39.8)	22.3 (39.0)	40.7 (43.5) ^#,§^	<0.001	0.44
BP	74.1 (9.2) ^†^	44.9 (26.2)	70.0 (20.9) ^#^	76.6 (27.8) ^#^	<0.001	1.21
GH	56.9 (8.1)	55.9 (21.1)	59.4 (23.5)	63.5 (26.2)	0.12	0.36
VT	60.7 (6.5) ^‡^	62.8 (17.7)	66.9 (18.8)	62.6 (17.6)	0.07	−0.01
SF	80.9 (5.9)	74.6 (25.7)	75.4 (31.6)	80.8 (29.0)	0.18	0.24
RE	71.0 (10.5) ^†,‡,&^	39.8 (46.3)	59.1 (45.8) ^#^	84.6 (30.2) ^#,§^	<0.001	0.96
MH	70.7 (3.8) ^†,&^	64.5 (18.3)	72.7 (18.2) ^#^	78.7 (18.2) ^#,§,†^	<0.001	0.78

PCS, physical component summary; MCS, mental component summary; PF, physical functioning; RP, role limitations due to physical health problems; BP, bodily pain; GH, general health; VT, vitality; SF, social functioning; RE, role limitation due to emotional problems; MH. mental health; ES, effect size (Glass’s delta, calculated between 6 months and pre-operation); ^#^, *p* < 0.001, compared with preoperation (THR PCS, *p* ≤ 0.005; THR/TKR MCS, *p* < 0.05); ^§^, *p* ≤ 0.003, compared with 6 weeks (THR MCS, *p* < 0.05); ^§,†^
*p* = 0.03, compared with 6 weeks; ^†^, *p* < 0.005, pre-operation compared with norm; ^‡^, *p* < 0.05, 6 weeks compared with norm; &, *p* < 0.03, 6 months compared with norm; *, *p =* 0.022 for patients with THR compared with patients with TKR; **, and *p =* 0.01 for patients with THR compared with patients with fracture.

**Table 3 healthcare-11-00329-t003:** The VAS and BAI at pre-operation, discharge, and 6-week and 6-month follow-up.

	Pre-Operation (*n* = 100)	Discharge (*n* = 100)	6 Weeks (*n* = 84)	6 Months (*n* = 55)	*p*	ES
	Mean (SD)					
VAS *	4.9 (3.0)	3.0 (2.5) ^†^	1.7 (2.3) ^†,#^	1.2 (2.5) ^†,#^	<0.001	1.23
THR	4.7 (3.1)	2.5 (2.3) ^†^	0.8 (1.5) ^†,#^	1.2 (2.7) ^†^	<0.001	1.13
TKR	5.1 (3.0)	3.2 (2.6) ^†^	2.2 (2.6)*^,†^	0.9 (1.8) ^†,#^	<0.001	1.40
Fracture	4.5 (3.0)	3.0 (2.8)	1.8 (2.4)	2.8 (4.3)	0.10	0.57
BAI	7.8 (6.7)	5.2 (4.6) ^†^	4.4 (5.9) ^†^	1.9 (3.7) ^†,#,§^	<0.001	0.88
THR	7.5 (6.2)	4.8 (5.1) ^†^	3.3 (4.8) ^†^	0.7 (1.3) ^†,#^	<0.001	1.10
TKR	7.2 (5.7)	4.8 (4.4) ^†^	5.3 (6.9)	3.0 (5.0) ^†^	0.017	0.74
Fracture	10.3 (10.0)	5.9 (4.4)	3.3 (3.9) ^†^	1.0 (0.8) ^†^	0.019	1.02

VAS, visual analogue scale; BAI, Beck Anxiety Inventory; ES, effect size (Glass’s delta, calculated between 6 months and pre-operation); ^†^, *p* ≤ 0.001, compared with pre-operation (THR/TKR/fracture BAI, *p* < 0.05); ^#^, *p* ≤ 0.001, compared with discharge (THR/TKR VAS *p* < 0.01, THR BAI, *p =* 0.003); ^§^, *p =* 0.01, compared with 6 weeks; *, and *p =* 0.046 patients with THR compared with patients with TKR.

**Table 4 healthcare-11-00329-t004:** The association of VAS and BAI with summary measures and subscales of the SF-36.

Summary Measure	VAS	BAI		VAS 6 Weeks	BAI 6 Weeks		VAS 6 Months	BAI 6 Months
Preoperation			6 weeks			6 months		
PCS	−0.34 ***	ns	PCS	−0.46 ***	−0.44 ***	PCS	−0.69 ***	−0.46 ***
MCS	ns	−0.43 ***	MCS	ns	−0.55 ***	MCS	−0.70 ***	−0.74 ***
Subscales								
PF	ns	ns	PF	−0.30 **	−0.37 ***	PF	−0.59 ***	−0.57 ***
RP	ns	ns	RP	ns	−0.24 **	RP	−0.56 ***	−0.34 **
BP	−0.45 ***	ns	BP	−0.62 ***	−0.45 ***	BP	−0.93 ***	−0.50 ***
GH	−0.37 ***	−0.35 ***	GH	−0.36 ***	−0.44 ***	GH	−0.61 ***	−0.49 ***
VT	−0.31 ***	−0.37 ***	VT	−0.43 ***	−0.59 ***	VT	−0.65 ***	−0.64 ***
SF	ns	ns	SF	ns	−0.45 ***	SF	−0.81 ***	−0.53 ***
RE	ns	ns	RE	ns	−0.32 **	RE	−0.74 ***	−0.59 ***
MH	−0.22 *	−0.44 ***	MH	−0.28 *	−0.62 ***	MH	−0.61 ***	−0.74 ***

VAS, visual analogue scale; BAI, Beck Anxiety Inventory; PCS, physical component summary; MCS, mental component summary; PF, physical functioning; RP, role limitations due to physical health problems; BP, bodily pain; GH, general health; VT, vitality; SF, social functioning; RE, role limitation due to emotional problems; MH, mental health; ns, not significant; *, *p* < 0.05; **, *p* < 0.01; and ***, *p* < 0.005.

## Data Availability

Not applicable.

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
