# Peer review of "Pre- and Post-Operative Education and Health-Related Quality of Life for Patients with Hip/Knee Replacement and Hip Fracture"

_healthcare, 2023, doi:10.3390/healthcare11030329_

Round 1

Reviewer 1 Report

This interesting article examined a group of Taiwanese patients receiving a variety of arthroscopic surgeries along with an educational intervention with tests at pre-test, release (discharge), and two post-times. Results are encouragingly positive. Strong points include the variety of testing instruments employed and the inclusion of not only one but two post-tests.

I would suggest some improvements that are relatively minor.

1. The authors employ many acronymics: for example for Health Quality there is HQOL, MOS, PCS, and MCS, not to mention the 8 subscales. If it would not betray the writing style that the authors consider appropriate, the article could be improved by the more frequent employment of phrases rather than acronymics.

2. The background data for arthroscopic surgeries covers many nations but not Taiwan. What (roughly) are local rates? Also is there socialized medicine in Taiwan or are the patients more likely to be paying for their own surgeries?

3. I am assuming (from the references cited) that the "effect size" referred to is Cohen's d. This should be specifically stated throughout the paper (including the abstract), as there are many different techniques for assessing effect size, some of them being directly associated with ANOVA (e.g., eta squared, omega squared).

4. How was the "poor" HQOL referred to in line 183 determined? I see the  means but not the method of labeling the HQOL as "poor" (although I am quite ready to believe that it was, indeed, poor).

5. What, exactly, are the test scores referred to in line 185? (A little more redundancy in discussing variables would make this communication clearer.)

6. Section 3.2, which contains the meat of the results on Health Quality of Life, is very crammed and abbreviated. It should probably be expanded somewhat and a figure might help the reader interpret the data. There is nothing like a descending line to show what PCS, MCS, VAS and BAI  are going across time.

7. The relative strength of correlations referred to in lines 208-218 was not tested for significance and so may be insignificant. It should not be mentioned. It would be perfectly acceptable to refer to "moderate" "medium strong" and "strong" correlations descriptively.

8. The authors could compare the various surgeries or groups of them in terms of scores (but this was not their original question, so not absolutely  necessary).

9. The total sample size should be mentioned in the Method.

10. The limitations should include mention of the lack of a control group, This was not a randomized trial design. Information it yields is still very informative, but there are some questions it cannot answer.

11. Which post-hoc analysis was employed after the ANOVA?

Reviewer 2 Report

This is a well-written account of an interesting study. In general, patient data are fairly well reported, but it was disappointing to see that no comparative information was presented. The study design did not include a control group, but since these patients came from a single hospital, it is reasonable to assume that average information for patients outside this study should be available (e.g. length of stay / age / gender / procedure). Such comparative data would at the very least assure the reader that the patients in the study were not atypical. Is there a compelling rationale for NOT including this?

The patient mix in the study population is identified, but thereafter there is no comparison of outcomes between the 3 sub-groups. At the very least (and as a condition of aggregating the patients into a single cohort) there needs to be a breakdown of patient characteristics at baseline - ideally then repeated at follow-up. Is there any differential between the sub-groups?

This manuscript is by no means alone in its reporting of SF-36, which is correctly identified as a profile measure. The current FDA advice regarding so-called Patient Reported Outcomes (PRO) data is that they should come directly from the patient. The SF-36 weighting system used to generate subscale scores masks the detail contained within items. The Bodily Pain subscale is a case in point, as it is based on only two items. Analysis of individual items within SF-36 may reveal ) that some items perform better than others, for example as prognostic indicators of post-intervention health status or HrQoL.

SF-36 contains 1 item (self-rated health status - Excellent / Very Good /  Good / Fair / Poor) which has been wholly overlooked within this study report. It forms part of what is labelled as "General Health" but is in fact the most powerful and widely used type of category rating scale used to assess health status. Most, perhaps all, population health surveys include such a rating scale. It would be far more productive to report this as a separate variable, independently of SF-36.

It is often the case that when mean (tail-whisker) VAS ratings are plotted against 5-point rating categories, highly significant differences are immediately evident. 

Was no exploration made (for example OLS regression) of the relationship between Beck / VAS at follow-up and SF-36 at baseline. This might be organised if SF-36 subscales were represented as z-scores compared with population norms and/or individual items were recoded to form dummy variables (0/1).  
